# The Clutter Simulation of a Known Terrain by the 3D Parabolic Equation and RCS Computation

**DOI:** 10.3390/s22197452

**Published:** 2022-09-30

**Authors:** Zhiyi Wang, Jieru Ding, Min Wang, Shuyuan Yang

**Affiliations:** 1National Laboratory of Radar Signal Processing, Xidian University, Xi’an 710071, China; 2Information and Education Technology Center, Xi’an University of Finance and Economics, Xi’an 710071, China; 3School of Artificial Intelligence, Xidian University, Xi’an 710126, China

**Keywords:** parabolic equation, ground clutter, simulation, RCS

## Abstract

Ground clutter data are usually generated using a statistical model, but they cannot effectively reflect the spatial distribution characteristics of ground objects. It is important in practical projects to effectively predict ground clutter when terrain data are known. In this paper, a scheme of ground clutter simulation based on the three-dimensional parabolic equation (3DPE) model is proposed. Radio wave propagation was modeled by the PE model and the spatial field distribution was solved. After the radar cross section (RCS) calculation based on space cells, the ground clutter information data were obtained. Then the radar echo was obtained and the clutter map was simulated. The simulation experimental results show that the ground clutter map simulated by the proposed method has a good reference value, meets the demand of strong clutter area prediction under known terrain conditions, and provides a theoretical basis for radar location and optimal deployment.

## 1. Introduction

The interference of environmental electromagnetic scattering on radar target detection is usually called clutter. The clutter modeling and simulation technologies help to better test the radar performance. For example, in order to detect drone swarms in urban areas with known terrain, it is necessary to predict the ideal location of radar erection through the simulation of ground clutter.

The classical clutter simulation methods, based on statistical models, the coherent clutter is obtained through methods such as Zero-Memory Non-Linear (ZMNL) or Spherically Invariant Random Processes (SIRP) [1,2,3,4,5]. However, echoes generated by these methods are not suitable for the ground clutter prediction under known terrain conditions and cannot help the erection and the deployment optimization of networked radar.

To meet the above application requirements, the radar cross section (RCS) calculation model was introduced in the early stage, but these methods lack the considerations of electromagnetic wave diffraction. Therefore, the radio wave propagation model has been introduced into the clutter simulation method in recent years. The main solution methods of radio wave propagation include numerical calculation, ray tracing, and parabolic equation (PE). In particular, PE has been widely studied because of its fast solution by step-by-step Fourier transform (SSFT) [6,7]. PE was initially applied to the deployment and location of mobile communication base stations. Meanwhile, the urban area was equivalent to a plane, the antenna beam was generally assumed to be flat and isotropic, and ideal results are then obtained. However, when PE is introduced into the radar field, there are two new problems: (1) how to solve the PE model in a three-dimensional space (3D) and (2) how to calculate the RCS of ground objects when the transceiver shares a single antenna.

On the one hand, to ensure that PE is effective in two-dimensional (2D) cases, it is necessary to assume that a scene is an infinite plane [8] or that the radar horizontal beam is extremely narrow, to avoid the problem of PE solution in 3D cases. In [3], the concept of the Gaussian linear antenna has been introduced to simplify the solution of the initial field in 3D space. Obviously, these methods are only suitable for solving specific 3D PE models. On the other hand, the bistatic model, without considering the scattering coefficient model and the solution of RCS, was assumed in [9] to solve easily the forward field in the PE model. The radar equation based on the PE is proposed to compute backscattering power in [10]. However, these two problems have not been satisfactorily solved in [3,7,8,9].

To solve the above problems, this paper develops a 3D split-step Fourier transform (3D-SSFT) method based on the 3DPE model and proposes a surface feature RCS calculation method based on the cells to generate the surface feature clutter of a known terrain. Furthermore, according to the radar echo model, the corresponding ground clutter signal and ground clutter map prediction results are obtained by simulation.

The remainder of the paper is organized as follows. Section 2 provides a brief overview of the simulation scheme and the short description of the radar echo model. The short description of PE model, the proposed 3D-SSFT algorithm, and approach of computation of RCS are shown in Section 3. In Section 4, we analyze the influence of buildings on 3D-SSFT and the influence of RCS computation on different sizes of cell and show the simulation results of clutter map. A summary of the paper is presented in Section 5.

## 2. The Simulation Scheme

### 2.1. The Radar Echo Model

Taking the radar whose transmitting waveform is a chirp signal as an example, regardless of RF modulation, the transmitting signal $S_{t}$ can be expressed as:(1)$$S_{t}(t)=P_{t}rect(\frac{t}{T})\exp(j\pi \mu t^{2})$$
where $P_{t}$ is the transmission power, rect(•) is the window function, the size of the window is determined by T, μ is the frequency modulation rate, and t is the time variable.

According to the radar equation, the received power $P_{r}=\frac{P_{t}G_{t}G_{r}\sigma }{4\pi \lambda^{2}r^{4}}$, where $G_{t}$ and $G_{r}$, as determined by the azimuth θ of a target, are the transmit and receive gain of the radar antenna, and $G_{t}=G_{r}=G(\theta )$. For N scatterers $\left\{{\mathrm{Point}}_{n},n=1,2,...,N\right\}$ with coordinate $\left(\theta_{n},r_{n},\varphi_{n}\right)$ and its radial velocity $v_{n}$ and RCS $\sigma_{n}$, the radar received echo $S_{r}^{n}$ can be expressed as follows:(2)$$S_{r}^{n}(t)=A\left(\theta_{n},r_{n},\varphi_{n},\sigma_{n}\right){S^{\prime }}_{t}(t;r_{n},v_{n})$$
where $A\left(\theta_{n},r_{n},\varphi_{n},\sigma_{n}\right)$ represents the complex envelope, and $S_{t}$ is transformed to ${S^{\prime }}_{t}$ after the Doppler modulation and the time-delay caused by $v_{n}$ and $r_{n}$, respectively. The radar echo ${\bar{S}}_{r}^{N}$ of N scatterers can be rewritten as:(3)$${\bar{S}}_{r}^{N}=\sum_{n=1}^{N}S_{r}^{n}$$

### 2.2. The Proposed Simulation Scheme

The workflow chart of the proposed simulation scheme is shown in Figure 1, where the internal module of generating scattering point information includes: 3D-SSFT of 3DPE model, the RCS calculation method based on cells. As in [6,8], the echoes are considered to be caused by isotropic scatterers which correspond to cells. Different from [3,8,11], we need the RCS of the cell rather than the distribution of the backward field, considering that the radar usually uses a single receiving and transmitting antenna and is erected from a single base. Therefore, this paper introduces an RCS calculation method based on cells. The module of generating scattering point information is the core of our simulation scheme whose output includes: coordinate (θ,r,φ), radial velocity $v_{r}$, and RCS σ of all cells.

Five main steps of our scheme include: (1) inputting radar parameters, known terrain data and its scattering coefficient, and parameters required by PE model; (2) solving the forward field by 3D-SSFT; (3) calculating the RCS of all cells; (4) generating the echoes of the known terrain data and objects from the radar–echo model; and (5) outputting the ground clutter map with flight targets.

## 3. The 3D Parabolic Equation and RCS Computation

### 3.1. The Parabolic Equation Model

Suppose the z-axis is set as the paraxial direction in 3D space and the time dependence of the fields is assumed as $e^{-j\varpi t}$, the reduced function is defined as u(x,y,z)=exp(−ikz)ψ(x,y,z), where k=2π/λ is the wavenumber, and ψ illustrates the components of time-harmonics either as an electric field or magnetic field.

The wave equations for each component of fields can be written as:(4)$$\frac{\partial^{2}u}{\partial x^{2}}+\frac{\partial^{2}u}{\partial y^{2}}+\frac{\partial^{2}u}{\partial z^{2}}+2ik\frac{\partial u}{\partial z}+k^{2}\left(n^{2}-1\right)u=0$$
where n is the refractive index of the medium. Considering $Q=\sqrt{\frac{1}{k^{2}}\frac{\partial^{2}}{\partial x^{2}}+\frac{1}{k^{2}}\frac{\partial^{2}}{\partial y^{2}}+n^{2}}$, (4) is simplified and decomposed.
(5)$$\frac{\partial u}{\partial z}=-ik(1-Q)u$$

The solutions to (5) correspond to the forward propagating waves, and (5) is called the standard parabolic equation model. Based on the 2D wide-angle (WA) SSFT [12], this paper proposes a 3DPE method, which includes the initial field solution of 3D Gaussian pattern, 3D absorption boundary, and 3D-SSFT method (3D-SSFT).

### 3.2. 3D-SSFT

This solution u(x,y,z) based on the 3DPE model can be used to calculate u(x,y,z+Δz) along *z* with the steps of Δz, once the initial field distribution and Boundary Conditions (BC) are given.

#### 3.2.1. The Initial Field

The initial field in the spatial domain can also be represented as u(x,y,0) at z=0. A Fourier transform pair is formed between the initial field u(x,y,0) and the specify Gaussian antenna pattern g(x,y), and its expression in 3D is:(6)$$\begin{array}{l} g(x,y)=\frac{\exp\left[-\frac{1}{2}\left(\frac{{\left(x-x_{S}\right)}^{2}}{\sigma_{1}^{2}}+\frac{{\left(y-y_{S}\right)}^{2}}{\sigma_{2}^{2}}\right)\right]}{2\pi \sigma_{1}\sigma_{2}} \end{array}$$
where $\left(x_{s},y_{s}\right)$ is the antenna’s position, and $\sigma_{1}$ and $\sigma_{2}$ represent the beamwidth along with x-axis and y-axis, respectively. The pattern in the wavenumber domain can be obtained:(7)$$g(k_{x},k_{y})=\exp\left[-\frac{k_{x}^{2}\omega_{x}^{2}}{4}-\frac{k_{y}^{2}\omega_{y}^{2}}{4}\right]$$
where $\omega_{\zeta }=\frac{2\ln2}{k\sin\left(\theta_{\zeta }/2\right)}$, $\theta_{\zeta }$ is 3 dB beamwidth of ζ-axis, and ζ represents x or y axis. When the elevation angle $\theta_{elv}$ and the azimuth angle $\theta_{azi}$ are known, and the pattern of the antenna can be denoted as $g(k_{x}-k\sin\theta_{elv},k_{y}-k\sin\theta_{azi})$, the initial field can also be represented using:(8)$$u(x,y,0)=\exp[ikx\sin\theta_{elv}+iky\sin\theta_{azi}-\frac{{\left(x-x_{s}\right)}^{2}}{\omega_{x}^{2}}-\frac{{\left(y-y_{s}\right)}^{2}}{\omega_{y}^{2}}]$$

#### 3.2.2. Boundary Conditions (BC)

The perfectly matched layer (PML) is used as an absorbing boundary condition to solve the PE model shown in Figure 2. This BC is achieved by applying windowing functions in order to eliminate reflection effects [13]. In this paper, we extend the Tukey window from one dimension to two dimensions, in which the 1D Tukey window w(ζ) on the ζ axis is defined as follows:(9)$$w(\zeta )=\left\{ \begin{array}{l} \frac{1}{2}\left\{1+\cos\left(\frac{2\pi }{R}[\zeta -R/2]\right)\right\},\;0\leq \zeta <\frac{R}{2}\\ 1,\;\;\;\;\;\;\;\;\;\;\;\;\frac{R}{2}\leq \zeta <1-\frac{R}{2}\\ \frac{1}{2}\left\{1+\cos\left(\frac{2\pi }{R}[\zeta -1+R/2]\right)\right\},1-\frac{R}{2}\leq \zeta \leq 1 \end{array}\right.$$
where R represents the maximum range in the ζ axis. The 2D Tukey window function is as follows:(10)$$w(x,y)=w(x)\times w{\left(y\right)}^{T}$$
where × represents Kronecker product and T represents transpose.

#### 3.2.3. 3D-SSFT

This paper proposes 3D-SSFT based on [12], and the specific step-by-step calculation formula is given by the following formula:(11)$$u(x,y,z+\Delta z)=F^{-1}\left\{C(k_{x}^{})C(k_{y}^{})F[u(x,y,z)]\right\}$$
where $C(k_{\zeta }^{})=\exp\left[-ik_{\zeta }^{2}\frac{\Delta z}{k_{}+\sqrt{k_{}^{2}-k_{\zeta }^{2}}}\right]$, $k_{\zeta }$ is the component of the wavenumber $k_{}$ on the ζ axis, Δz is step along z, and F and $F^{-1}$ represent 2D Fourier transform and inverse transform, respectively. The field strength is composed of both the forward and backward components, as seen in [14,15,16].

### 3.3. Computation of RCS Based on the Cell

The RCS of a target is usually estimated by physical optics and geometric optics. In [17], RCS is calculated by the finite difference time domain method (FDTD). Reference [8] demonstrated that the forward field can be obtained by solving the PE model, and then the RCS of the target in the bistatic radar can be calculated. The research in [8] has attracted a lot of attention, such as [6,7,18,19]. In this paper, ROI is divided into non-overlapping cells, each of which is a $\tilde{x}$ × $\tilde{y}$ × $\tilde{z}$ cube. The RCS of each cell are defined as:(12)$$\sigma (\theta ,\varphi )=\underset{r\to \infty }{\lim}4\pi r^{2}{\left|\frac{u^{s}(x,y,z)}{u^{i}(x,y,z)}\right|}^{2}$$
where θ and φ represent azimuth and elevation, respectively; r denotes the distance between radar and cell; and $u^{s}$ and $u^{i}$ express scattered field strength and incident field strength, respectively. Then we have:


(13)
x=rcosθ,y=rsinθcosφ,z=rsinθsinφ


The RCS of cells is given as [7,8]:(14)$$\sigma (\theta ,\varphi )=\frac{k^{2}\cos^{2}\theta }{\pi }{\left|\int_{-\infty }^{+\infty }\int_{-\infty }^{+\infty }u_{s}\left(x_{0},y^{\prime },z^{\prime }\right)e^{-ik\sin\theta \left(y^{\prime }\cos\varphi +z^{\prime }\sin\varphi \right)}dy^{\prime }dz^{\prime }\right|}^{2}$$

The proposed scheme is summarized in the following pseudo-code shown in Algorithm 1.
  **Algorithm 1****:** The pseudo-code of our scheme.
Input: Radar Parameters, Known Terrain Data Output: the Clutter Map1:Initialize $S_{t}$, the azimuth of beam $\bar{\theta }=0$, $G(\bar{\theta })$, w, u(x,y,0).2:**while**$\bar{\theta }\leq 360$3: **for**
$z\in [0,z_{\max}]$ **do**4:  z=z+Δz;5:  $u(x,y,z+\Delta z)=F^{-1}\left\{C(k_{x}^{})C(k_{y}^{})F[u(x,y,z)]\right\}$;6:  according to reflection coefficients, $u_{s}$ is calculated by u;7:  $\sigma (\theta ,\varphi )=\frac{k^{2}\cos^{2}\theta }{\pi }{\left|\int_{-\infty }^{+\infty }\int_{-\infty }^{+\infty }u_{s}\left(x_{0},y^{\prime },z^{\prime }\right)e^{-ik\sin\theta \left(y^{\prime }\cos\varphi +z^{\prime }\sin\varphi \right)}dy^{\prime }dz^{\prime }\right|}^{2}$;8: **end for** where $\left(\sigma_{n}\right)$ is the nonzero subset of σ in all space cells9: **for**
n∈[1,N]
**do**10:  $S_{r}^{n}(t)=A\left(\theta_{n},r_{n},\varphi_{n},\sigma_{n}\right){S^{\prime }}_{t}(t;r_{n},v_{n})$;11: **end for**12: ${\bar{S}}_{r}^{N}=\sum_{n=1}^{N}S_{r}^{n}$;13: as the antenna scans $\bar{\theta }$ increases;14:**end while**15:after pulse compression to obtain the clutter map from ${\bar{S}}_{r}^{N}$

## 4. Simulation Results and Discussion

### 4.1. The Influence of Buildings on 3D-SSFT

The main parameters of the radar in the experiment are as follows: the carrier frequency is 9.35 GHz, the pulse repetition frequency PRF is 40 KHz, and the bandwidth B is 20 MHz. The vertical and horizontal beamwidths are set to 10° and 1°, respectively, and the surface scattering coefficient is assumed to be 0.1.

The first experiment is to show the influence of ground objects on radio wave propagation. The radar is at *x* = 50 m, *y* = 100 m, and *z* = 0 m. The heights of the three buildings are 30 m, 50 m, and 80 m, respectively. The width and thickness of the buildings are 110 m and 100 m, respectively. The cell is 8 m × 1 m × 20 m and the ROI is 3000, 100, and 200 m on the *z*, *x*, and *y*-axis, respectively. The forward field strength distributions on different *x*–*z* planes are obtained by 3D-SSFT. The field strength distributions on *y* = 100 m, 120 m, and 140 m are shown in Figure 3.

The field distribution can be seen from Figure 3: the field strength at the *y* = 100 m plane, where the radar is located, is the largest; and it gradually decreases with the increase of *z* in Figure 3b. However, in Figure 3c,d, under the influence of antenna directivity, the maximum field strength is not at the position where *z* is the smallest; in Figure 3b,c, under the influence of building shielding, the field strength at the back of the building changes significantly, and behind the building, with the increase of *z*, the electromagnetic wave diffusion effect is displayed.

### 4.2. Comparisons with Different Sizes of Cell

In the second experiment, in order to verify the feasibility of modeling ground objects and targets as cuboids, three cube targets with side lengths of 1 m, 10 m, and 100 m are set to compare the RCS results. The three targets are located at the same center, 1000 m away and 50 m high.

When the target side length is 1 m, one non-zero RCS value is −319.13 dBm. In the 10 m cube experiment, 10 cells of non-zero RCS value are obtained, as shown in Table 1. In the case of the 100 m cube, 9896 non-zero RCS values of cells are obtained. The statistical results are shown in the first column of Table 2.

In addition, in order to compare the difference between a combination of cubes and a single cube, we set 1000 cubes with side length of 10 m to obtain one 100 m cube combination with the equivalent size and position to a single 100 m cube and also obtain 9896 RCS statistical results, as shown in the second column of Table 2.

It can be seen from the experiments of three different size targets that small cubes lead to fewer non-zero RCS cells, and the larger the cube, the more non-zero RCS cells are obtained, which is consistent with the actual situation. The statistical results of RCS values generated by small cubes, which are equivalent to a large cube, and by a single cube are the same in Table 2. This result shows that it is feasible to equivocate large objects with small cubes.

### 4.3. Clutter Maps and Simulation Results

The digital elevation map in this paper is the urban elevation map formed by the superposition of the topographic height map and the building height map of a city provided by a research institute in China, as shown in Figure 4. One pixel corresponds to 1 m × 1 m area. The map covers a 3000 × 3000 m^2^ area, with altitudes between 450 and 700 m. We set the size of cells is 1 m × 1 m × 1 m to obtain 3000 × 3000 × 700 cells. The target is set to a 3 m × 9 m × 8 m cuboid with a scattering coefficient of 0.1, a circular flight path with a radius of 1000 m and an altitude of 600 m.

The clutter maps in dB of the proposed scheme are compared at different radar erection heights of 500, 550, 600, and 650 m, respectively, as shown in Figure 5. In Figure 5b–d The target track can be easily observed at the ring with a radius of 1 km. The target track is more obvious and the signal-to-clutter ratio is higher when the radar erection height is close to the target flight height. This shows that the radar erection height or beam pointing to the target has an important impact on the radar target detection.

In order to demonstrate the advantages of the proposed 3DPE in fidelity, 2D PE is used to generate range profiles (RP) from 0° to 360° azimuth, and its results are shown in Figure 6b. It can be seen from the area in black boxes that RP in Figure 6b is more distinct in the adjacent azimuth than that of 3DPE in Figure 6a. The reason is that 2D PE does not consider the beamwidth of radar, which does not meet the needs of real projects.

## 5. Conclusions

This paper verifies and develops the ground clutter simulation method based on PE to predict the ground clutter when the terrain data are known. The proposed method can intuitively predict the coverage of radar detection airspace and the strong ground clutter area, meet the needs of strong clutter area prediction under known terrain conditions, and provide a theoretical basis for radar location and optimal deployment.

## Figures and Tables

**Figure 1 sensors-22-07452-f001:**
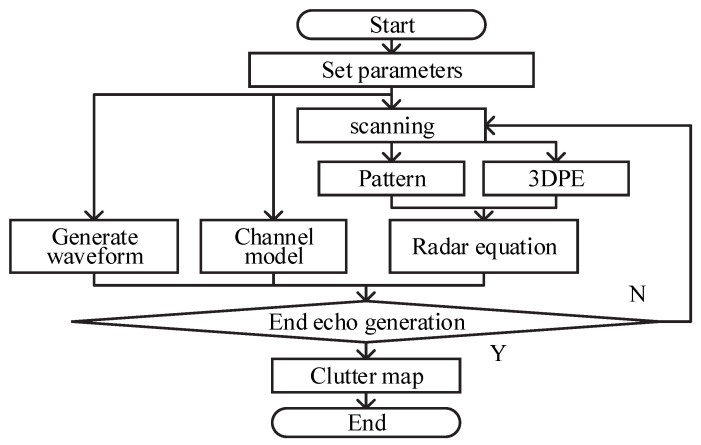
The workflow chart of the proposed simulation scheme.

**Figure 2 sensors-22-07452-f002:**
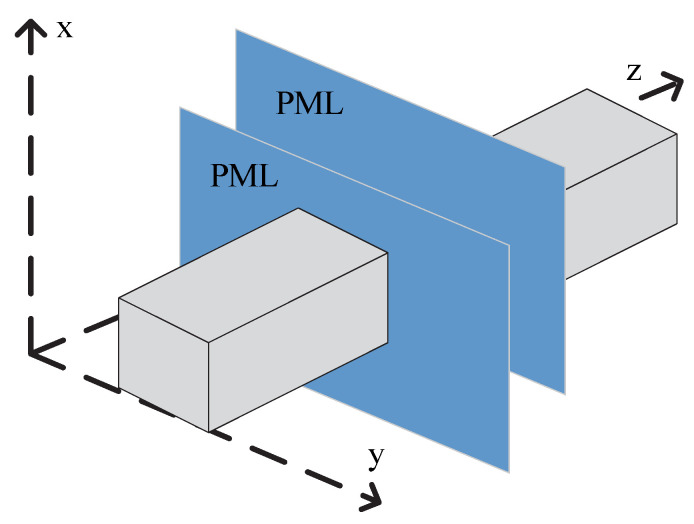
The ROI (gray) and the PML (blue).

**Figure 3 sensors-22-07452-f003:**
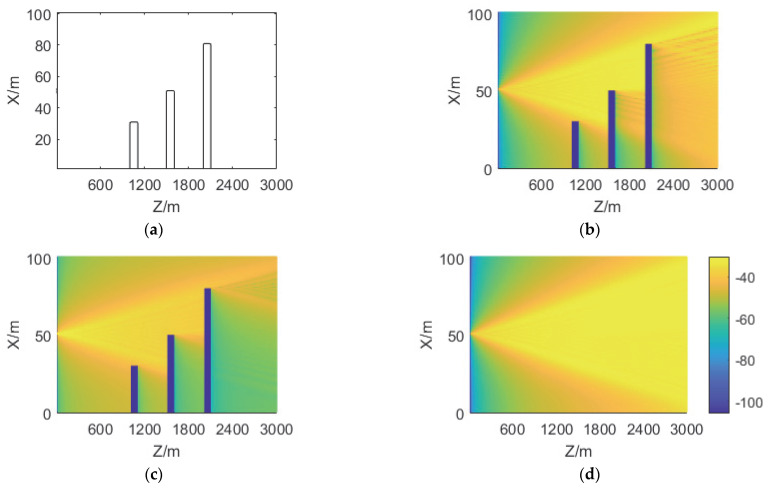
Forward field distributions in dB on different *x*–*z* planes. (**a**) shows the locations of the three buildings. (**b**–**d**) are field distributions of *y* = 100 m, 120 m, and 140 m, respectively. It should be pointed out that the farther away from the plane where the radar is, the weaker the shelter effect of the building gradually becomes.

**Figure 4 sensors-22-07452-f004:**
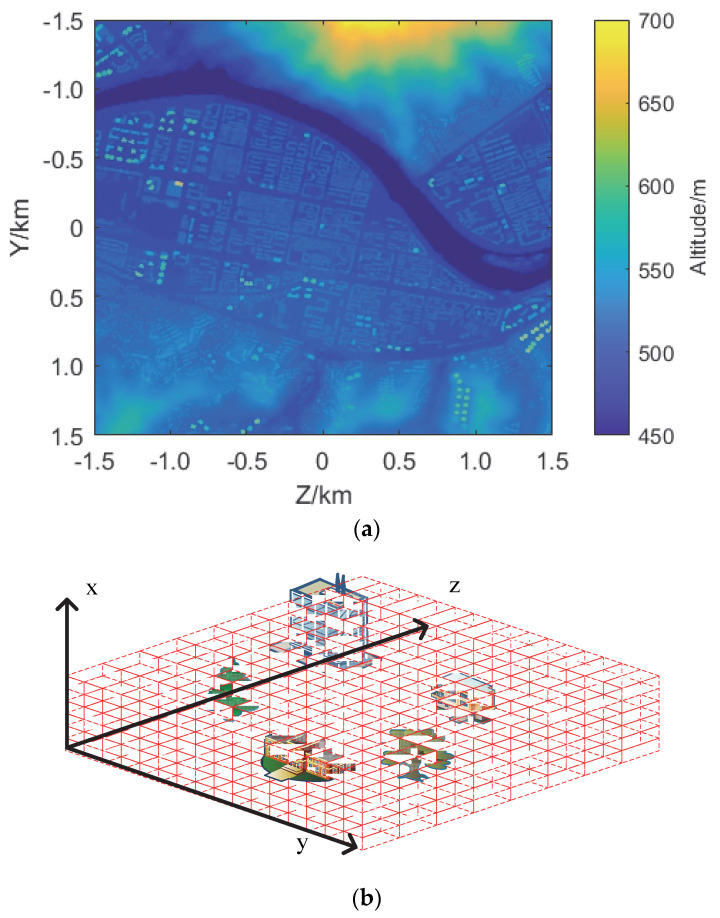
The urban elevation map in (**a**) with altitudes between 450 and 700 m. This map is divided into space cells as shown in (**b**).

**Figure 5 sensors-22-07452-f005:**
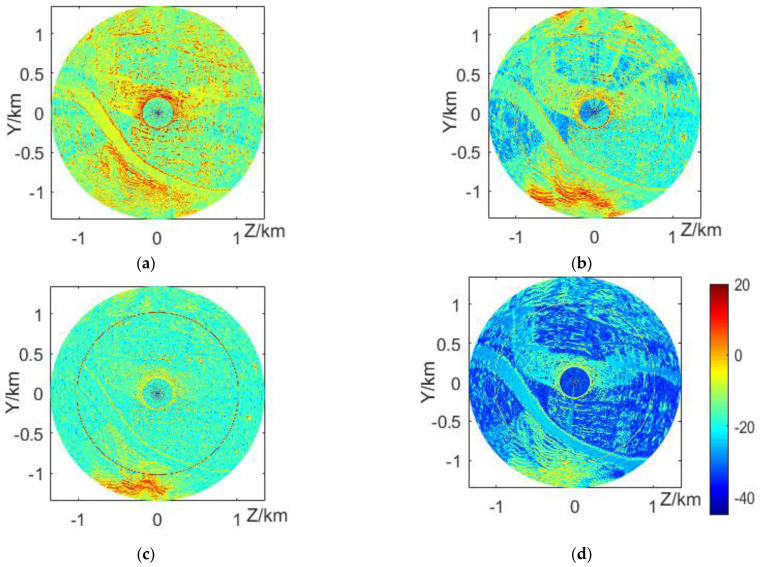
The clutter maps with at different radar altitudes in (**a**–**d**) of 500 m, 550 m, 600 m, and 650 m, respectively. These results correspond to the DEM, in which the mountainous area at the lower side of the map corresponds to the red strong clutter area at the lower part of the clutter map, the flat terrain and river area on the left form the blue weak clutter area, and there are obvious red spots in the clutter map corresponding to some high-rise buildings on the right. The “hollow” phenomenon in the central area of the clutter map is mainly due to the failure to consider the influence of the near-field effect and radar sidelobe.

**Figure 6 sensors-22-07452-f006:**
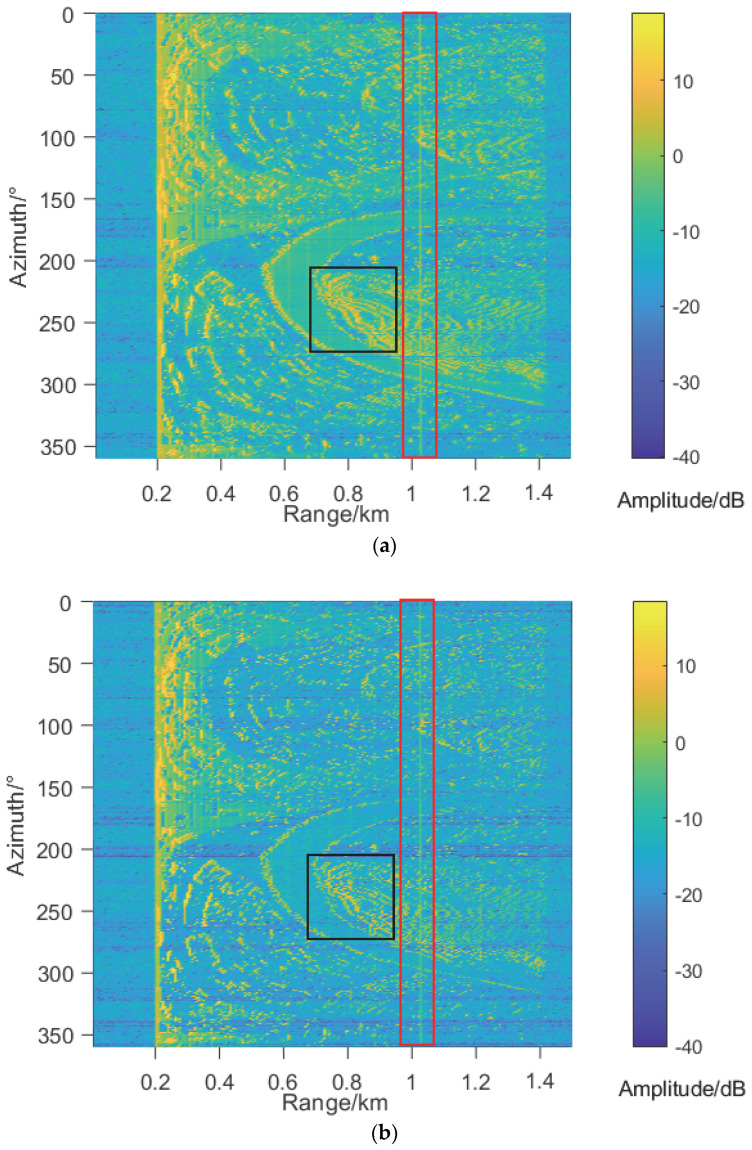
The comparison of range profiles at different azimuth angles between 3D PE in (**a**) and 2D PE model in (**b**) under the same conditions as the above and radar erection heights is 500 m. The target is marked by a red box.

**Table 1 sensors-22-07452-t001:** The comparison with different sizes of cubes.

Num	RCS/dBm	Num	RCS/dBm
1	−251.27	6	−235.04
2	−247.48	7	−241.65
3	−250.04	8	−246.30
4	−255.91	9	−241.37
5	−252.25	10	−242.60

**Table 2 sensors-22-07452-t002:** The comparison between single cube and cube combination.

RCS/dBm	A 100 m Cube	1000 Cubes
>−248	45	45
−248~−265	108	108
−265~−282	169	169
−282~−299	401	401
−299~−316	364	364
−316~−333	380	380
−333~−350	1960	1960
−350~−368	3800	3800
−368~−385	2140	2140
<−385	529	529

## Data Availability

The data presented in this study are available on request from the corresponding author. The data are not publicly available due to project restrictions.

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
