# Peer review of "The Clutter Simulation of a Known Terrain by the 3D Parabolic Equation and RCS Computation"

_sensors, 2022, doi:10.3390/s22197452_

Round 1
Reviewer 1 Report
This paper proposes a clutter simulation method of a known terrain for radar. Authors extend the Parabolic Equation (PE) model into Three-Dimension Parabolic Equation (3DPE) model using 3D split-step Fourier Transform (3D-SSFT) and calculate Radar Cross Section (RCS) based on 3D cells to generate the surface feature clutter of a known terrain with known scattering coefficient. After that, echos of the known terrain data and objects are generated from the radar-echo model and the ground clutter map is accordingly depicted.
This paper is well written, but the following problems should be considered before acceptance:
1, The workflow chart of figure 1 is too brief to represent your core contributions of your method. It would be better to change it into architecture of the proposed method with figures and brief text description to visualize your method.
2, in 3.2.2, the parameter r in equation 9 is not defined, please check the equation and further explain why you can directly extend Tukey window into 2D with w(x) and w(y) using Kronecker product.
3, It would be better to compare your method with other clutter simulation method in experiment to show the novelty and advantage.
4, The reference format of this article is not proper. Please check your Latex grammar and use the correct reference format. The same as the “3000×3000m2” in 4.3.
5, Although you explain abbreviations such as RCS and PE in the end of this article, it would be better to explain it when it first appears in your paper.
Author Response
Response to Reviewer 1 Comments
Point 1: The workflow chart of figure 1 is too brief to represent your core contributions of your method. It would be better to change it into architecture of the proposed method with figures and brief text description to visualize your method.
Response 1: My response for Point 1. (in red)
The additional pseudo code for the simulation scheme is presented in Section 3.
Point 2: 2, in 3.2.2, the parameter r in equation 9 is not defined, please check the equation and further explain why you can directly extend Tukey window into 2D with w(x) and w(y) using Kronecker product.
Response 2: My response for Point 2. (in red)
Thank the reviewer for pointing out this problem. We have modified equation 9 and added parameter definition. The Boundary Conditions are achieved by applying windowing functions in order to eliminate reflection effects. The extension directly using Kronecker product does not affect the step-by-step solution of the region of interest.
Point 3: 3, It would be better to compare your method with other clutter simulation method in experiment to show the novelty and advantage.
Response 3: My response for Point 3. (in red)
This is a very helpful suggestion, and we have added some simulation results and the comparison with 2D PE model in section 4.
Point 4: 4, The reference format of this article is not proper. Please check your Latex grammar and use the correct reference format. The same as the “3000×3000m2” in 4.3.
Response 4: My response for Point 4. (in red)
We have corrected it according to the comment of the reviewer.
Point 5: 5, Although you explain abbreviations such as RCS and PE in the end of this article, it would be better to explain it when it first appears in your paper.
Response 5: My response for Point 5. (in red)
We have corrected it according to the comment of the reviewer.

Reviewer 2 Report
In this manuscript, a simulation for a clutter of a known terrain by the 3D parabolic equation and RCS computation is proposed. However, the paper lack of present a significance for a research paper with a poor technical quality. The authors are recommended to improve the introduction part. It is not clear what existing issues of conventional methods are and how to solve these issues in the current work. It would be better if the contents referring with key points of the simulation and further technical details can be explained more thoroughly in the corresponding sections of the manuscript.
In this same context, section 4 is just a simple description of a simulation procedure, some definitions in this section are not relevant for a research presentation and lack originality and innovation which indeed does not reflect a qualitative contribution. It is not clear if the proposed method is merely a procedure to setup a simulation workflow with only one controlled scenario.
The structure of the paper can be significantly improved and reformulated. Furthermore, for some specific sections, such as section 2 which describes the simulation scheme it is not clear whether parameters have been calculated without or/pseudo code; if yes, please could be included for the simulation scheme and its practical implementation for the specific scenario in order to support the description of the methodology proposed and addressed in the prediction, and theoretical results.
The results and discussion, as well as the conclusion section can be presented in a more punctual way to highlight the main contributions.
Author Response
Response to Reviewer 1 Comments
Point 1: In this manuscript, a simulation for a clutter of a known terrain by the 3D parabolic equation and RCS computation is proposed. However, the paper lack of present a significance for a research paper with a poor technical quality. The authors are recommended to improve the introduction part. It is not clear what existing issues of conventional methods are and how to solve these issues in the current work. It would be better if the contents referring with key points of the simulation and further technical details can be explained more thoroughly in the corresponding sections of the manuscript.
Response 1: My response for Point 1. (in red)
We have improved the introduction part. This paper verifies and develops the ground clutter simulation method based on PE. As the reviewer said, The purpose of this paper is how to solve these issues in the current work, and there is not a conventional method. The contents referring with key points of the simulation and further technical details have be expanded.
Point 2: In this same context, section 4 is just a simple description of a simulation procedure, some definitions in this section are not relevant for a research presentation and lack originality and innovation which indeed does not reflect a qualitative contribution. It is not clear if the proposed method is merely a procedure to setup a simulation workflow with only one controlled scenario.
Response 2: My response for Point 2. (in red)
We have added some simulation results and the comparison with 2D PE model in section 4.
Point 3: The structure of the paper can be significantly improved and reformulated. Furthermore, for some specific sections, such as section 2 which describes the simulation scheme it is not clear whether parameters have been calculated without or/pseudo code; if yes, please could be included for the simulation scheme and its practical implementation for the specific scenario in order to support the description of the methodology proposed and addressed in the prediction, and theoretical results.
Response 2: My response for Point 3. (in red)
There are the description for the effect of some parameters in section 4, such as size of cells and radar altitudes, which are not calculated, but set according to the actual situation. And a pseudo code for the simulation scheme is presented in Section 3.
Point 4: The results and discussion, as well as the conclusion section can be presented in a more punctual way to highlight the main contributions.
Response 2: My response for Point 4. (in red)
This is a very helpful suggestion, and we have made corresponding improvements in this paper.

Round 2
Reviewer 2 Report
The author's responses in this revised version show a significantly improved presentation concerning the punctual recommendations suggested in the previous version.